# Machine learning models of healthcare expenditures predicting mortality: A cohort study of spousal bereaved Danish individuals

Alexandros Katsiferis[1,2]*, Samir Bhatt[1,3], Laust Hvas Mortensen[1,2], Swapnil Mishra[1,4], Majken Karoline Jensen[1,2], Rudi G. J. Westendorp[1]

**1** Section for Epidemiology, Department of Public Health, University of Copenhagen, Copenhagen, Denmark, **2** Statistics Denmark, Denmark, **3** Department of Infectious Disease Epidemiology, Imperial College London, London, United Kingdom, **4** Saw Swee Hock School of Public Health, National University of Singapore, Singapore, Singapore

* alexandros.katsiferis@sund.ku.dk

## Abstract

### Background

The ability to accurately predict survival in older adults is crucial as it guides clinical decision making. The added value of using health care usage for predicting mortality remains unexplored. The aim of this study was to investigate if temporal patterns of healthcare expenditures, can improve the predictive performance for mortality, in spousal bereaved older adults, next to other widely used sociodemographic variables.

### Methods

This is a population-based cohort study of 48,944 Danish citizens 65 years of age and older suffering bereavement within 2013–2016. Individuals were followed from date of spousal loss until death from all causes or 31st of December 2016, whichever came first. Healthcare expenditures were available on weekly basis for each person during the follow-up and used as predictors for mortality risk in Extreme Gradient Boosting models. The extent to which medical spending trajectories improved mortality predictions compared to models with sociodemographics, was assessed with respect to discrimination (AUC), overall prediction error (Brier score), calibration, and clinical benefit (decision curve analysis).

### Results

The AUC of age and sex for mortality the year after spousal loss was 70.8% [95% CI 68.8, 72.8]. The addition of sociodemographic variables led to an increase of AUC ranging from 0.9% to 3.1% but did not significantly reduce the overall prediction error. The AUC of the model combining the variables above plus medical spending usage was 80.8% [79.3, 82.4] also exhibiting smaller Brier score and better calibration. Overall, patterns of healthcare expenditures improved mortality predictions the most, also exhibiting the highest clinical benefit among the rest of the models.

**Data Availability Statement:** The current study uses detailed data on individuals. This means that even if direct identifiers like name, date of birth and

street address are removed from the data, it is still possible to re-identify the individuals in the study, which would breach basic principles of data protection. Consequently, the data can only be shared under specific conditions. According to Danish law, scientific organizations can be authorized to work with data within Statistics Denmark and can provide access to individual scientists inside and outside of Denmark. Data are available via the Research Service Department at Statistics Denmark: www.dst.dk/da/TilSalg/Forskningsservice for researchers who meet the criteria for access to confidential data.

**Funding:** This work is supported by The Novo Nordisk Foundation (https://novonordiskfonden.dk/) Challenge Programme for the project Harnessing the Power of Big Data to Address the Societal Challenge of Aging (NNF17OC0027812). The funders had no role in study design, data collection and analysis, decision to publish, or preparation of the manuscript.

**Competing interests:** LHM is employed at Statistics Denmark, the national Danish Statistics office. This does not alter our adherence to PLOS ONE policies on sharing data and materials.

## Conclusion

Temporal patterns of medical spending have the potential to significantly improve our assessment on who is at high risk of dying after suffering spousal loss. The proposed methodology can assist in a more efficient risk profiling and prognosis of bereaved individuals.

## Introduction

The ability to accurately predict survival is pivotal for clinical decision-making, facilitating the timing and choice of medical interventions, contributing to a more precise counseling of patients with regards to their prognosis [1]. The need for accurate predictions is even more relevant in old age, as individuals become frail, a phenomenon attributed to the ageing process [2]. Ageing inevitably comes with gradual molecular, tissue or organ specific damage, which results in older adults being less resilient to stressors, as the body is incapable to maintain homeostasis and regulate those mechanisms accounting for their adaptive capacity [3]. Older adults are also exhibiting a higher degree of heterogeneity than younger ones with regards to their health status [4], which further complicates to make a correct prognosis, also alluding to the fact that there is a need for further development of models that account for that. Nevertheless, researchers have investigated numerous, clinical, and physiological hallmarks for the purpose of a more accurate prognostication for older adults [5–7].

The majority of studies related to development of models for prognosis of survival have focused on using single time-point measurements of variables to predict survival thereafter. While availability of repeated measurements might not always exist due to time and financial constraints, previous research has stated that longitudinal studies could bring new insights into the understanding and predicting of health-related outcomes [8]. Recent studies demonstrated increased prediction performance of clinical outcomes when using repeatedly measured variables related to human health and extracting Dynamical Indicators of Resilience (DIORs) thereof using a complex dynamical systems approach [9–12].

The aim of the current research is to determine the incremental value of various types of DIORs of healthcare expenditures to predict all-cause mortality in spousal bereaved adults 65 years of age and older. Spousal bereavement is major life event, especially relevant in the older adults, associated with increased mortality and hospitalization risk [13–16]. Thereby, we consider the development of a prognostic model, a useful tool for clinicians in identifying those individuals in high risk soon after spousal loss. Previous research has provided evidence for sex differences in mortality after spousal bereavement [17]. For that purpose, both pooled and sex-stratified prognostic models of mortality were developed. We categorized the indicators based on the type of signal they capture and assessed their predictive accuracy when implemented into the statistical models. We further developed another modelling strategy, by using the whole time series of expenditures without prespecifying any DIORs summarizing the latter. We evaluated those with respect to discrimination, overall prediction error, calibration, as well as clinical benefit. The main objective of our work is to investigate if and which of the temporal dynamics of healthcare costs, can improve predictive performance next to other widely used sociodemographic variables.

## Materials & methods

### Study population

The current study is based on data from national population registers available from Statistics Denmark (https://www.dst.dk), the central authority responsible for maintaining high quality

Danish registers and performing statistics on various aspects of life, such as socioeconomic and biomedical conditions of Danish citizens across time [18]. Each Danish citizen has a unique person identifier, which can be linked to registers. Specifically, our initial population consisted of all the Danish citizens 65 years of age and older (N = 933,970) who had been resident in Denmark for at least 5 years before their respective date of death. The follow-up period spanned up to 6 years, from January 1st, 2011 to December 31st, 2016 and during this period information on all types of healthcare expenditures (measured in thousands of Danish Kroners) had been collected in weekly intervals. Specifically, expenditures related to prescription drugs, hospitalization, primary, home, and residential care were all available. The register sources of the aforestated data and their respective computations are described elsewhere [19]. For the purposes of our analysis, we aggregated per week all the different expenditures into one variable. Thus, we consider the expenditures data as a read-out of an individual's health-status trajectory, given the fact that the Danish healthcare system is tax-based, and individuals have no incentive to refrain from medical spending due to out-of-pocket expenses. Apart from healthcare expenditures, we used information regarding the date of birth and death, sex, date of spouse's death, immigration status, number of comorbidities and number of children at baseline, all being virtually complete. We also used the affluence index at baseline as a measure of socioeconomic status. The aforementioned index is computed for each individual of a specific age and at a specific year (2011), by aggregating their preceding year's wealth and income, with the latter being multiplied by a capitalization factor K for retirement income. Each person in the study was then assigned to a cohort-adjusted percentile. The exact computation, methodology, and previous applications of the index are described elsewhere [19, 20].

To compare prognostic models of all-cause mortality risk, we restricted the initial population of 933,970 older adults to a cohort of those who suffered spousal bereavement during the time window 2013–2016 (N = 50,245) and analyzed time series of healthcare expenditures up to two years before bereavement. From those 50,245 individuals, we excluded some 1,248 for whom we could not compute any meaningful time series features due to constantly zero healthcare expenses, hence, the final sample size of our cohort was 48,944 persons. All were followed up from date of spousal loss until death from all causes or the end of the first year after bereavement whichever came first.

## Healthcare expenditures and knowledge discovery

For each individual, we computed various DIORs, summarizing different properties of the underlying dynamic process. Specifically, we summarized each time series based on three distinct properties. An illustration of all indicators used is presented in Fig 1. First, we computed DIORs of overall trend and status for each individual. These indicators revolve around the average of expenditures and their rate of change across time. Second, we computed indicators based on dispersion properties. Such metrics aim to capture aspects of variance and entropy of a time series. Third, we calculated indicators related to the memory properties, which are computed based on lags of the time series and aim to reveal temporal correlations amongst time-intervals. The predictive performance of these categories was compared by implementing them separately and jointly in prognostic models. Their mathematical formulas are not presented in the current study but are described elsewhere [21, 22].

## Statistical methods

We used the XGBoost (eXtreme Gradient Boosting) tree-based algorithm to create our prognostic models. The latter relies on the boosting ensemble technique, a process in which decision trees are sequentially added to the ensemble to minimize the residuals (errors) of the

**Dispersion Dynamics**

Mean Squared Error
Largest mean shift in sliding windows
Number of Crossing Points
Spikiness
Permutation Entropy
Non-zero squared coefficient of variation
Variance of tiled means (stability)
Variance of tiled variances(lumpiness)
Time of largest variance shift
Time of largest distributional shift
Time of largest mean shift

**Memory Dynamics**

Lag-1 autocorrelation of original, differenced
and detrended time series
Longest Flat Spot
Hurst coefficient
Sum of squares of the first ten autocorrelation
coefficients of original, (double) differenced and
detrended time series
Partial autocorrelation of original, differenced and
detrended time series
Sum of squares of the first ten partial
autocorrelation coefficients of original, (double)
differenced and detrended time series.

**Overall Trend and Status Dynamics**

Average
Linearity Coefficient
Trend Strength
Zero running mean
Zero starting proportion

**Fig 1. List of all DIORs of healthcare expenditures time series used in prediction models by signal category.**

predictions of the previously fitted ones [23]. We fitted various models by starting with simple ones including only age and sex as predictors, then proceeded by creating our reference (benchmark) model which implemented age, sex, and other sociodemographic variables (affluence index, number of comorbidities, immigration status, number of children). To assess the predictive performance of the different signal categories of DIORs, we fitted additional models, each one of them implementing the predictors of the benchmark plus a set of DIORs describing specific dynamics respectively.

The hyperparameters of the XGBoost (number and maximum depth of trees, learning rate, number of randomly selected predictors, minimal node size, and number of iterations before stopping) were tuned for each model based on a range of 30 different candidate values per

parameter in a 5-fold cross-validated dataset of the initial training set (75% of the full dataset). The grid of the candidate values was constructed using a space-filling, maximum entropy design, with the goal of finding a configuration of points that covers the parameters space with the smallest chance of overlapping or redundant values [24]. Finally, the combination of parameters that minimized the logarithmic loss of the model was selected. The prognostic value of the models was assessed by comparing their c-statistic (AUC), as a measure of discrimination ability i.e., how well predictions differentiated individuals who died with those who did not. The 95% confidence intervals for the AUC are constructed based on the standard errors of the metric, obtained with an estimate of the influence function, and automatically extracted via the R package "riskRegression" [25, 26].

We validated the generalization ability of our models internally using a holdout set (25% of the full dataset). For each model, we compared the observed proportion of deaths within the year after bereavement with the predicted number of events and depicted them using calibration plots [27]. We evaluated the overall prediction error using the Brier score, a measure of assessing a model's goodness of fit, with smaller scores indicating better performance. For a given individual, the Brier Score is the average of the squared difference between the person's predicted risk and their observed binary outcome. In a dataset, we first apply our model to extract all the predicted risks of all individuals and then compare these to the observed outcomes. The score is then computed by averaging all the squared differences between the risks and the outcomes, thus being the mean squared error of the prediction. The Brier Score is also a proper scoring rule, takes values between zero and one, and assesses both the discrimination and calibration ability of a model, thus being an appropriate metric to compare rival models in terms of performance [26, 27]. The standard error of the Brier Score is computed as the square root of the standard deviation of the squared residuals, followed by Wald-type 95% confidence intervals [25, 26].

To assess the potential clinical impact of those models we performed decision curve analysis, the latter reporting the proportion of accurately diagnosing and treating bereaved individuals ("Net Benefit") for various risk thresholds of the models compared with reference scenarios of either treating everyone or none [28–30]. Here, we define the Net Benefit as the proportion of individuals who are correctly classified as high risk, i.e., dying within the year after losing their spouse (True Positives) minus the proportion of individuals who are incorrectly classified as high risk (False Positives), the latter weighted by a factor related to the relative benefits and harms of each (odds of the risk threshold). We performed the decision curve analysis after fitting the models into the training (75% of total data) set and testing their utility in the holdout (25% of total data) set.

The R code describing the packages used and the data analysis performed in the current project is available in https://github.com/alkat19/Healthcare_Expenditures_Analysis.

### Ethics statement

All methods were performed in accordance with the relevant guidelines and regulations. Danish legislation allows for register-based research of this type to be conducted without the consent of participants and without ethical committee approval. The study was conducted according to the rules of the Danish Data Protection Agency. All data was held at Statistics Denmark, which is the Danish national statistical institution.

### Results

The overall study sample consisted of 48,944 individuals with an average age of 78.5 ± 6.4 (SD) years at the date of bereavement. Of the sample, 32,224 (65.8%) were females, with 96.5% of

the participants being Danish, and 66.6% having two children or more. Based on the descriptive statistics, males had a higher socioeconomic status (28.1% being placed in the top quartile of affluence index) compared with females (17.4% in the top quartile). Males had on average higher number of comorbidities. During the 1-year of follow-up, approximately 8.0% of males (N = 1355) and 4.0% of females (N = 1399) died. The information regarding the distribution of the socio-demographic variables across the validation and holdout sets used in the current analysis is presented in Table 1, for males and females, respectively. While we did not perform a stratified validation-holdout split on the outcome (mortality status), the distribution of mortality was comparable across the two datasets, i.e., 5.5% and 5.9%, respectively.

We initially performed an analysis with sex as a predictor variable in our models. The results of the analysis are illustrated in Fig 2 as bar plots. For comparison purposes, we fitted models with the predictors i) age and sex, ii) age + sex + various sociodemographics (benchmark), iii) benchmark + four basic dynamical indicators, iv) benchmark + specific category of dynamics, v) benchmark + all categories of dynamics. For each model, we showed the AUC and the Brier score of the models tested in the holdout set.

We observed a small increase in the discriminative ability (AUC [95% Confidence Interval (CI)]: 72.8 [71.0, 74.7], Brier: 5.3 [5.0, 5.7]) when adding the sociodemographics next to a model with only age and sex as predictors (AUC: 70.8 [68.8, 72.8], Brier: 5.4 [5.0, 5.7]). The

**Table 1. Distribution of socio-demographic variables across validation and holdout study samples.**

| | Validation Set | | P-value | Holdout Set | | P-value |
|---|---|---|---|---|---|---|
| | **Males** | **Females** | | **Males** | **Females** | |
| Sample Size | 12,593 (34.3%) | 24,115 (65.7%) | | 4,127 (33.7%) | 8109 (66.3%) | |
| Age at Bereavement | 79.0 (6.8) | 78 (6.1) | < 0.001 | 79.5 (6.78) | 78.1 (6.11) | < 0.001 |
| Immigration Status | | | | | | |
| Danish | 12,213 (97.0%) | 23,178 (96.1%) | < 0.001 | 4,013 (97.2) | 7,799 (96.2%) | |
| Immigrant | 380 (3.0%) | 937 (3.9%) | | 114 (2.8%) | 310 (3.8%) | |
| Affluence Index Group | | | | | | |
| 1$^{st}$ Quartile | 2,334 (18.5%) | 8,313 (34.5%) | < 0.001 | 791 (19.2%) | 2,822 (34.8%) | < 0.001 |
| 2$^{nd}$ Quartile | 3,129 (24.8%) | 5,966 (24.7%) | | 1,068 (25.9%) | 1,944 (24.0%) | |
| 3$^{rd}$ Quartile | 3,551 (28.2%) | 5,681 (23.6%) | | 1,152 (27.9%) | 1,898 (23.4%) | |
| 4$^{th}$ Quartile | 3,579 (28.4%) | 4,155 (17.2%) | | 1,116 (27.0%) | 1,445 (17.8%) | |
| Number of Comorbidities | | | | | | |
| Zero | 4,406 (35.0%) | 8,936 (37.1%) | < 0.001 | 1,479 (35.8%) | 2983 (36.8) | 0.37 |
| One | 3,204 (25.4%) | 6,339 (26.3%) | | 1,054 (25.5%) | 2,118 (26.1%) | |
| Two | 2,086 (16.6%) | 3,822 (15.8%) | | 662 (16.0%) | 1,306 (16.1%) | |
| Three | 1,322 (10.5%) | 2,327 (9.6%) | | 453 (11.0%) | 830 (10.2%) | |
| Four or More | 1,575 (12.5%) | 2,691 (11.2%) | | 479 (11.6%) | 872 (10.8%) | |
| Number of Children | | | | | | |
| Zero | 1,797(14.3%) | 2,732 (11.3%) | < 0.001 | 581 (14.1%) | 900 (11.1%) | < 0.001 |
| One | 2,709 (21.5%) | 5,092 (21.1%) | | 880 (21.3%) | 1,662 (20.5%) | |
| Two | 4,703 (37.3%) | 9,390 (38.9%) | | 1,566 (37.9%) | 3,233 (39.9%) | |
| Three | 2,443 (19.4%) | 4,881 (20.2%) | | 783 (19.0%) | 1,627 (20.1%) | |
| Four or More | 941 (7.5%) | 2,020 (8.4%) | | 317 (7.7%) | 687 (8.5%) | |

Data are mean (standard deviation) for continuous variables and n (%) for categorical ones

SD = Standard Deviation

P-values indicate statistically significant difference in the variables across the sex subgroups.

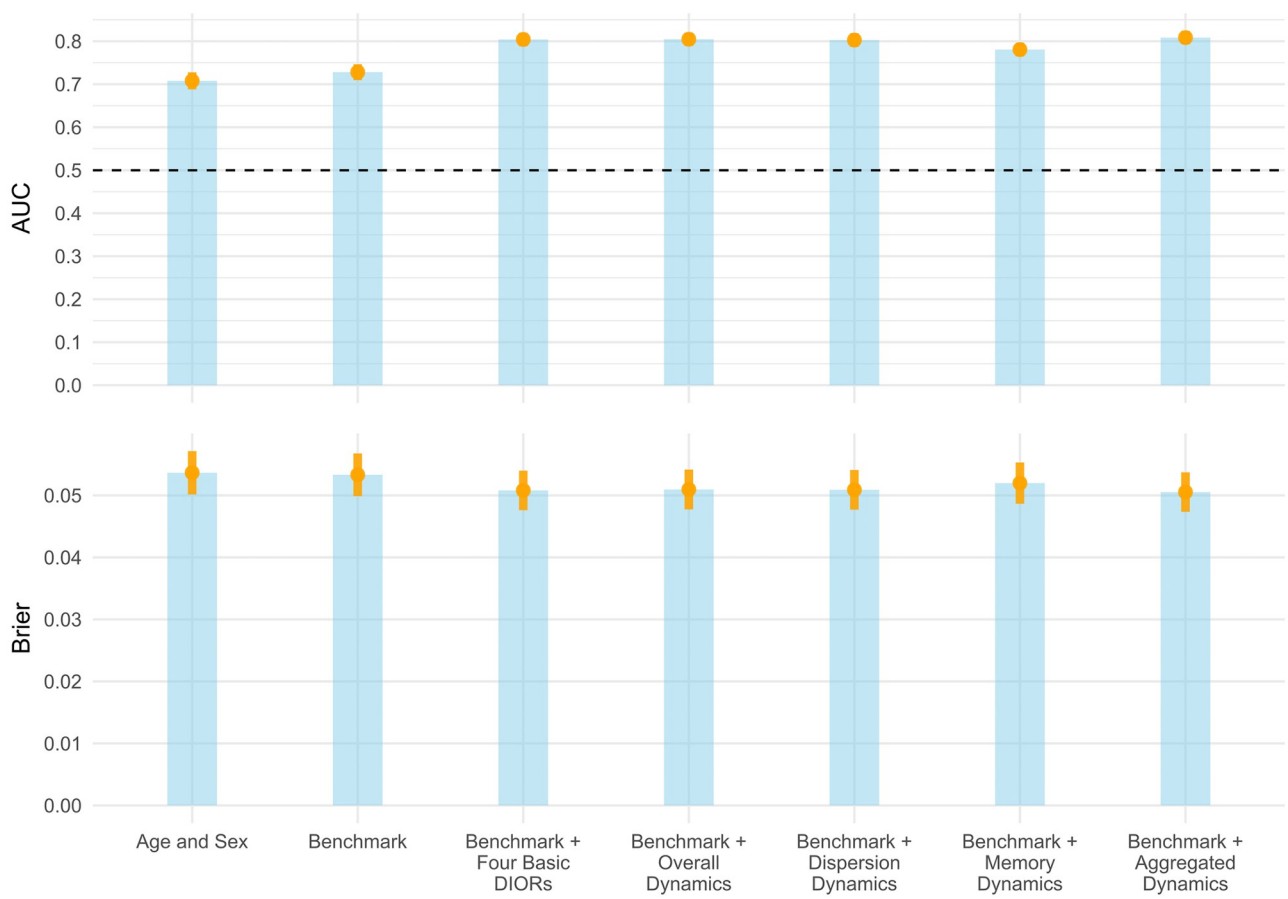

**Fig 2. Performance of risk prediction models for the non-stratified analysis.** Orange point-ranges specify performance measure value along with their 95% confidence intervals for the holdout set. The vertical dashed black line specifies the value of the AUC (0.5) for which a model is not able to discriminate.

addition of dynamical indicators to the benchmark model improved discrimination and reduced the overall prediction error, irrespective of what type of indicators were implemented. For example, the model with Benchmark + Aggregated Dynamics reached an AUC of 80.8 [79.3, 82.4], with the Brier score being 5.1 [4.7, 5.4]. We repeated the same analysis separately for males and females. The results, presented in Table 2, exhibited the same patterns as the non-stratified analysis, with the overall prediction performance being higher in the females dataset.

The 95% confidence intervals for the difference in the AUC and Brier Score of the models against the Benchmark one, are provided separately as S1 Table.

We evaluated the performance separately for different age groups to investigate if there are differences across the strata. The model 'Benchmark + Aggregated Dynamics' achieved higher performance (AUC: 78.6 [73.4, 83.7] and Brier: 2.1[1.7, 2.5]) in the youngest group, decreasing with age (Table 3).

We further assessed the variable importance of the models by computing the loss in the AUC after permuting each predictor (replacing its original value with random one for all observations in the dataset), repeated 1000 times. The results are illustrated in Fig 3, indicating that age and sex are the most important variables for the prediction (causing the biggest loss in

**Table 2. Performance of prediction models for 1-year mortality risk stratified on sex in the holdout set.**

| | Males | | Females | |
|---|---|---|---|---|
| | **AUC (%)** | **Brier (%)** | **AUC (%)** | **Brier (%)** |
| Age Only | 68.2 [65.0, 71.4]* | 7.0 [6.4, 7.7]* | 67.7 [64.9, 70.6]* | 4.2 [3.8, 4.6] |
| Age + Sociodemographics (Benchmark) | 71.5 [68.7, 74.4] | 6.9 [6.3, 7.6] | 72.1 [69.6, 74.7] | 4.1 [3.7, 4.5] |
| Benchmark + 4 Basic Dynamical Indicators | 78.3 [75.8, 80.8]* | 6.6 [6.1, 7.2]* | 81.5 [79.4, 83.6]* | 4.0 [3.6, 4.3]* |
| Benchmark + Overall Trend Dynamics | 77.6 [75.1, 80.1]* | 6.7 [6.1, 7.3]* | 81.9 [79.8, 83.9]* | 3.9 [3.6, 4.3]* |
| Benchmark + Dispersion Dynamics | 77.0 [74.4, 79.5]* | 6.7 [6.1, 7.3]* | 81.4 [79.3, 83.5]* | 4.0 [3.6, 4.3]* |
| Benchmark + Memory Dynamics | 74.6 [72.0, 77.2]* | 6.9 [6.2, 7.5] | 78.3 [76.1, 80.4]* | 4.1 [3.7, 4.4]* |
| Benchmark + Aggregated Dynamics | 77.5 [75.0, 80.1]* | 6.7 [6.1, 7.3]* | 81.9 [79.8, 83.9]* | 4.0 [3.6, 4.3]* |

Sociodemographics: Affluence Index, Number of comorbidities, Immigration type, Number of children

Basic Dynamical Indicators = Average and Linearity (Slope), Variance & Autocorrelation of detrended (residual) expenditures data

Aggregated Dynamics = Overall Trend + Dispersion + Memory Dynamics

AUC: Area Under Curve

Brackets show the 95% confidence intervals

The * next to the performance measures indicates statistically significant difference in performance (p-value < 0.05) compared to the Benchmark model, used as a reference for the contrasts.

the AUC of models when permuted), followed by the average medical spending of individuals, then the rest of dynamics and finally the sociodemographics.

The calibration ability of the pooled (non-stratified on sex) models, is illustrated in Fig 4. The two models, which implement DIORs, appear to be better calibrated than the rest, showing good calibration for risks below approximately 30%. Specifically, the models that use either age and sex only or age + sex + sociodemographics (benchmark) overestimate largely than the other two ones, for risks higher than 25%. Nonetheless, only few observations with risk above 50% existed and most of them were below 25%. The calibration plots, separately for males and females, are available in the S1 & S2 Figs.

The potential clinical benefit of the predictive models was evaluated via decision curve analysis. Fig 5 shows the Net Benefit of the models predicting all-cause mortality within the year after spousal bereavement, under various scenarios in the holdout set. Up to a risk threshold of 0.2, all the predictive models have a positive proportion of accurately and treated individuals, exceeding that of the reference scenarios, of either treating everyone or none. For risk thresholds 0.3 and higher, the models had an approximately even number of true and false positives, implying no added clinical utility. Specifically, across the threshold

**Table 3. Performance of "Benchmark + Aggregated Dynamics" model stratified on age groups.**

| Performance in holdout set for 1-year mortality risk | | | | |
|---|---|---|---|---|
| **Age Group** | **AUC (%)** | **Delta AUC (%)** | **Brier (%)** | **Delta Brier (%)** |
| 65–74 | 78.6 [73.4, 83.7]* | 28.60 [23.40, 33.70] | 2.1 [1.7, 2.5] | -0.04 [-0.13, 0.05] |
| 75–84 | 77.5 [74.8, 80.3]* | 27.52 [24.75, 30.28] | 4.3 [3.9, 4.7]* | -0.30 [-0.42, -0.18] |
| 85plus | 72.7 [69.8, 75.7]* | 22.72 [25.66, 19.77] | 10.4 [9.5, 11.3]* | -0.894 [-1.16, -0.62] |

AUC: Area Under Curve

Delta AUC = AUC of Model with dynamical indicators—AUC of a Model with no predictors (only intercept)

The * indicates statistically significant difference (p-value < 0.05) in performance of the models within each age group against a model with no predictors (intercept only)

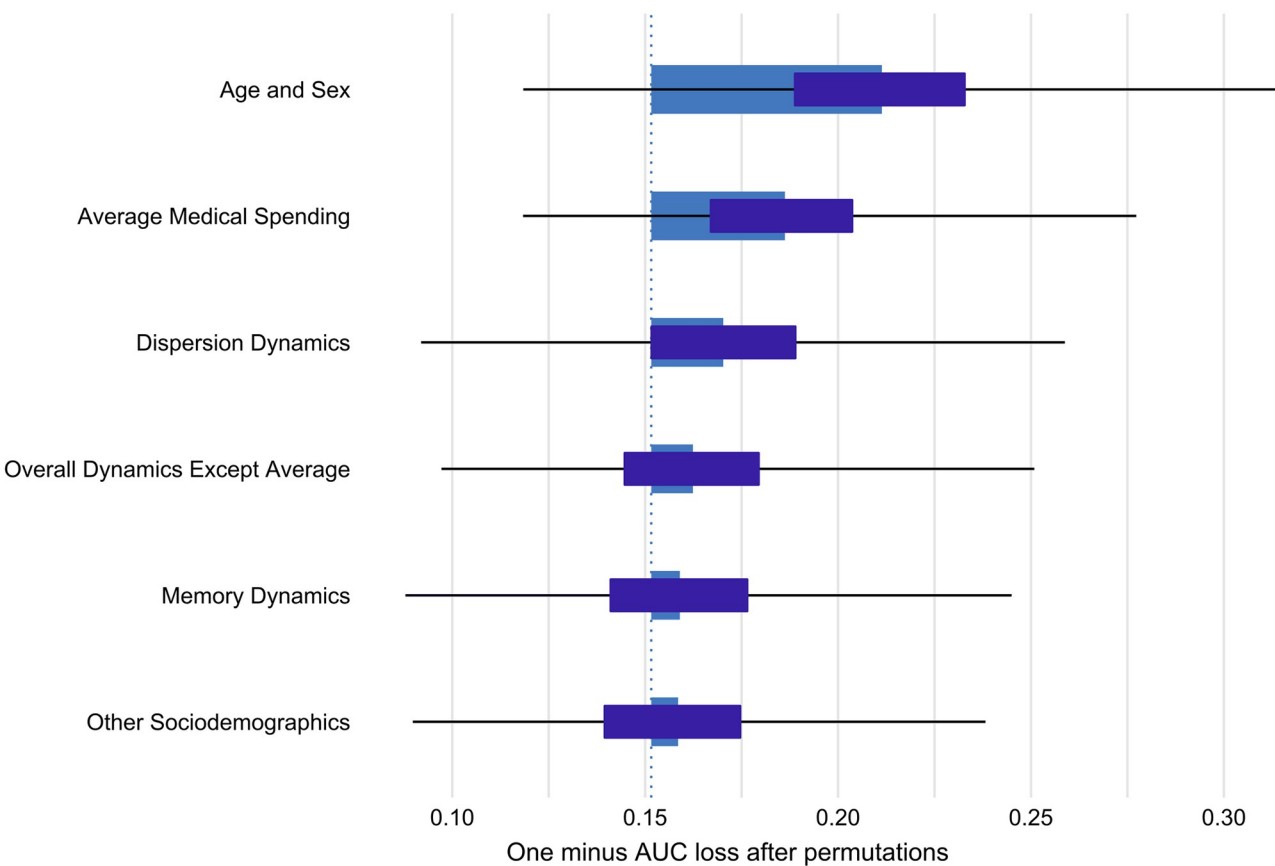

**Fig 3. Mean variable-importance calculated by using 1000 permutations and the one minus AUC loss function for the XGBoost model.** The bars in the plot indicate the mean values of the variable-importance measures for all explanatory variables. Box plots (dark blue) are added to the bars to provide an overview about the distribution of the values of the measure across the permutations.

range of 0 through 0.2, the models which make use of DIORs manifested higher Net Benefit than the model using only sociodemographics, however, their respective intervals overlapping with each other. Among the models with indicators, the Net Benefit at the chosen threshold of 0.1 was 0.019 [95% CI: 0.016, 0.023] for the 'All DIORs + Sociodemographics' full model. Consequently, in a scenario in which the full prediction model would be applied to 10,000 bereaved individuals, it would lead to 190 [160, 230] net, identified true positive cases. In terms of false negatives, the full model ("All DIORs + Sociodemographics") displayed lower false negative rates (FNRs) for thresholds up to 0.4 when compared against a model that did not use DIORs (S3 Fig).

Furthermore, we tried a different modelling strategy in order to further explore the predictive potential of healthcare expenditures data. Hence, instead of extracting DIORs, this time we fitted an XGBoost model using the raw expenditures computed per week as predictors, plus age and sex. Specifically, the raw expenditures were modelled by implementing 104 different predictors (equal to the number of weeks in a period of two years), each of them depicting the overall expenditures for each week, starting two years before spousal loss, and ending in the last week before the loss. The performance of the aforestated model (AUC: 80.8 [79.2, 82.4], Brier: 5.0 [4.7, 5.4]) was indistinguishable with the 'Benchmark + Aggregated Dynamics', with regards to all prediction metrics. With respect to variable importance of the model, age

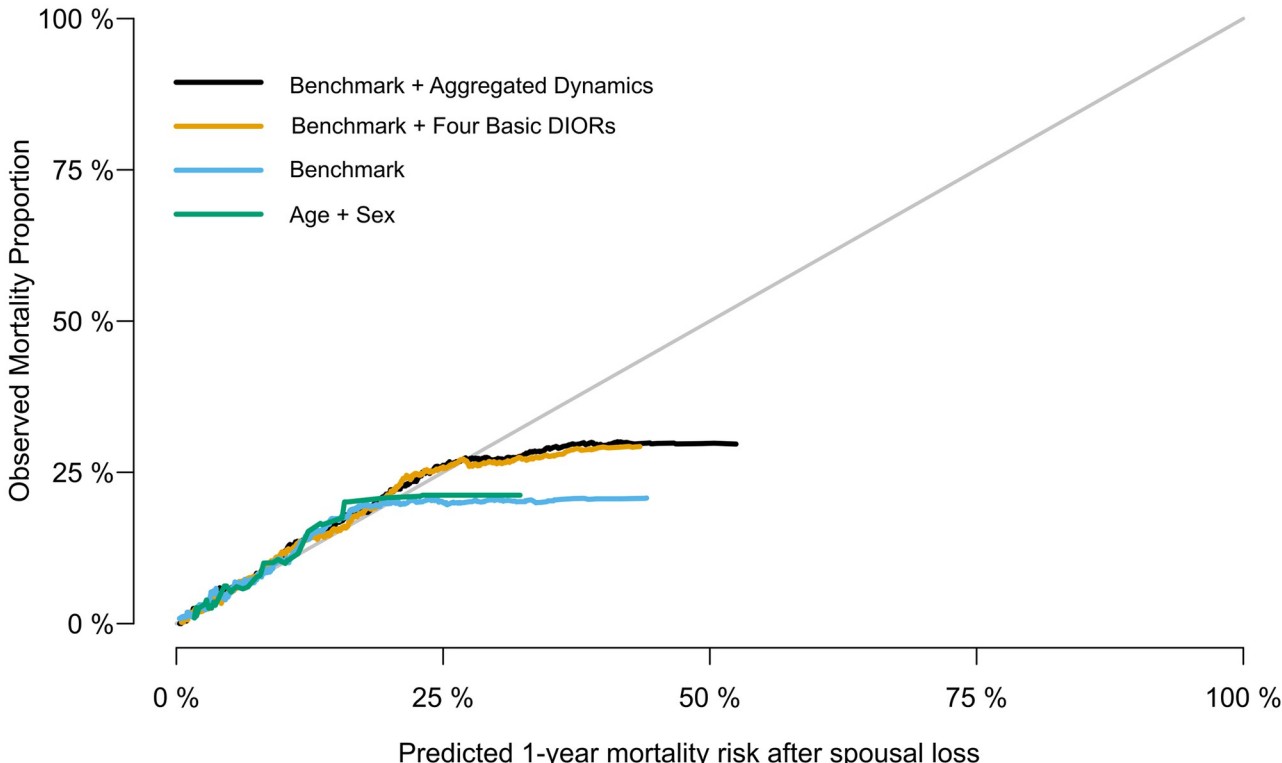

**Fig 4. Calibration plot showing risk estimates of all-cause mortality within the year after spousal bereavement against outcome proportions observed in the holdout dataset.** The plot (density type) displays the calibration of the XGBoost model fitted to the validation dataset. The model's risk predictions for the individuals in the hold-out dataset are ordered from low to high and shown in the x-axis ("Predicted 1-year mortality risk after spousal loss"). A fixed bandwidth defines how many hold-out dataset individuals are nearest neighbors to any given probability risk-value $p$ on the x-axis. The y-axis ("Observed Mortality Proportion") shows the relative frequency of the holdout set individuals with the outcome (Mortality) in the neighborhood around the value $p$. Points falling in the diagonal line represent perfect calibration of the model.

and sex were still the most important predictors, followed by the variable measuring the expenses spent during the last week before spousal bereavement. The expenditures measured in intervals more distant from bereavement, had a smaller impact on prediction. Last, we fitted a regularized logistic regression model in order to compare its performance with XGBoost. While prediction performance of the former (AUC: 80.3 [78.6, 81.9], Brier: 5.1[4.7, 5.4]) was similar with the latter, XGBoost appeared to be slightly more well-calibrated (Fig 6).

## Discussion

The current study evaluated the prognostic ability of dynamical indicators of resilience to predict all-cause mortality from time series of healthcare expenditures in a study sample of Danish older adults who had suffered spousal bereavement. The analysis produced the following observations. First, there was a slight increase in the discrimination ability, as measured by the AUC, by the addition of sociodemographic variables (affluence index, number of comorbidities, immigration type, and number of children) to a model with age and sex. However, the overall prediction error, indicating both discrimination and calibration, did not appear to be smaller, neither in the pooled nor in the stratified-on sex analysis, indicating that the added predictive value of commonly used sociodemographics was at best minimal. Second, while different dynamics (overall trend, dispersion, aggregate) of medical

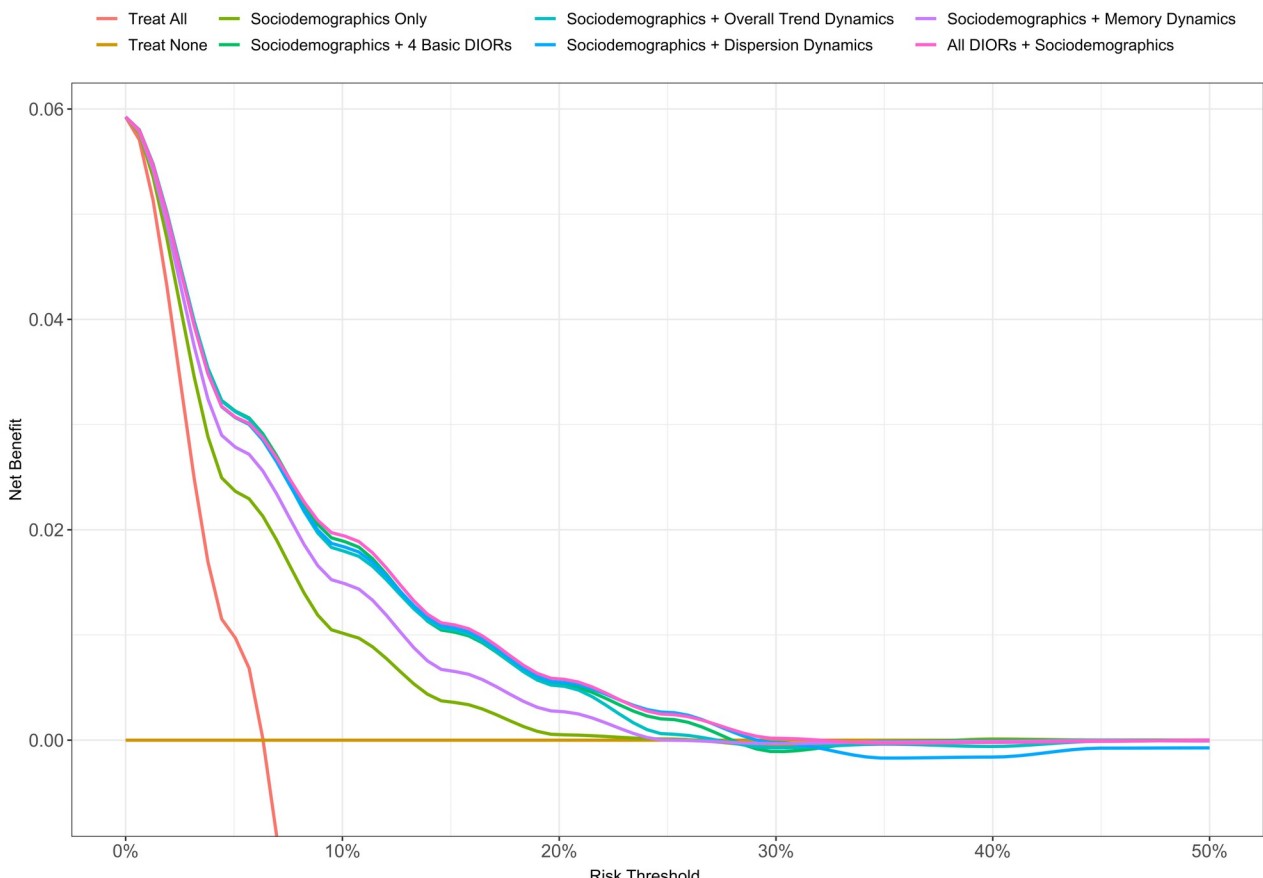

**Fig 5. Decision curve for validation of prediction model developed to estimate the risk of all-cause mortality with the year after spousal bereavement.** The x-axis ('Risk Threshold') shows the range of risk threshold probabilities, that is the probabilities which when exceeded, individuals are classified as high risk of dying within 1-year. The y-axis ('Net Benefit') shows the proportion, in True Positives, of accurately diagnosed and treated individuals, for each given threshold probability, after subtracting the weighted False Positives. A proportion of 0.02 implies 2 true positives for every 100 persons in the target population, without unnecessarily intervening on those. The red line represents the scenario of treating every individual 65 years of age or older who suffers bereavement without adhering to a predictive model ('Treat All'). The brown, flat line depicts the scenario of not intervening at all in the newly bereaved individuals ('Treat None'). The rest of the colored lines indicate the number of true positives (after subtracting the weighted False Positives) as a proportion based on the predictions of a specified model.

spending usage ended up predicting similarly, they all improved prognostic power when added to the models, which was apparent in all measures of performance, i.e., discrimination, calibration, and clinical utility. Third, the predictive potential for survival prognosis of health-related time series was superior to that of other static, cross-sectional variables describing the cohort participants, irrespective of the strategy chosen for the modelling of healthcare expenditures.

Our findings confirm those from previous studies where age and sex appeared to be the most important predictors for mortality risk. Similar with previous literature, while social, demographic and comorbidity differences might slightly advance predictive accuracy, our analysis showed that their performance based on age and sex alone appeared to be limited [5, 6, 31]. The accuracy of our prognostic tools for predicting mortality was progressively decreasing with age, a finding already observed in past studies, and potentially explained by the heterogeneity the oldest manifest with respect to their health [32, 33]. Nevertheless, we found evidence that utilizing longitudinal measurements of healthcare expenditures, as proxy of

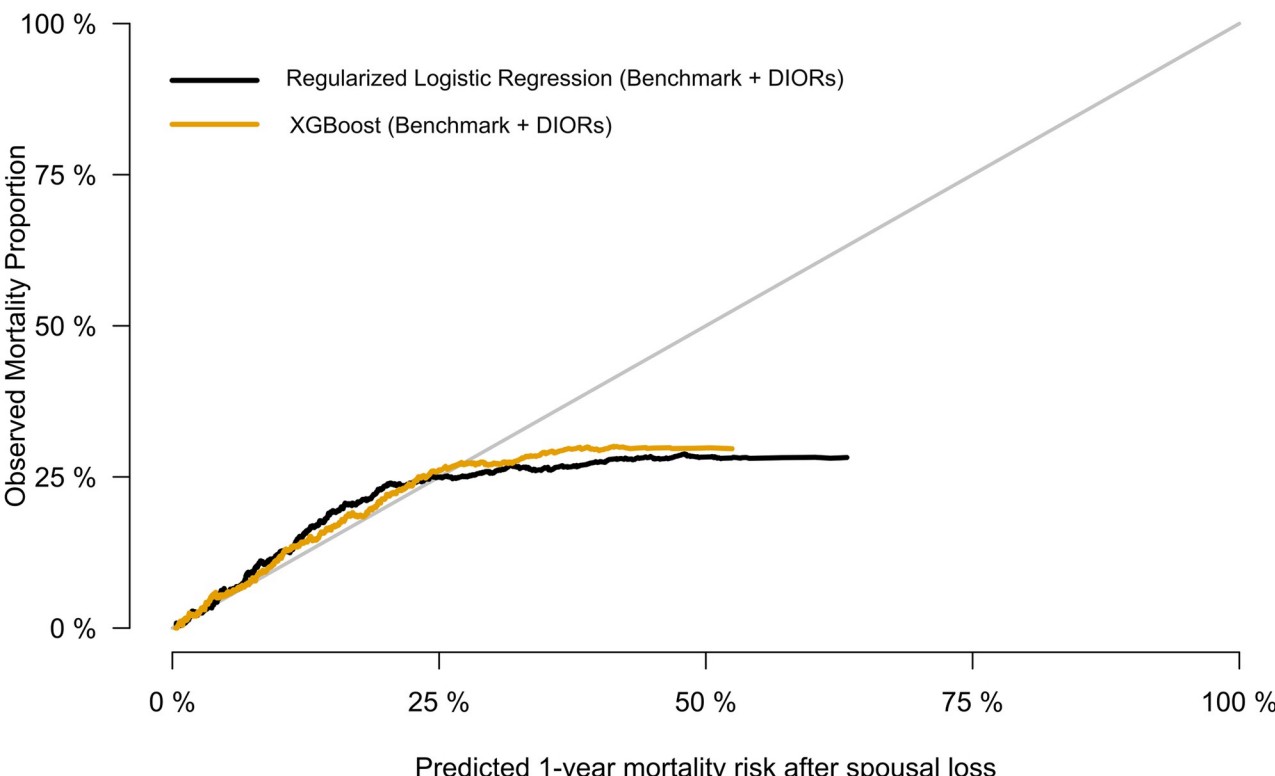

**Fig 6. Calibration plot for two different modelling algorithms (logistic regression and XGBoost).** Risk estimates are being shown of all-cause mortality within the year after spousal bereavement against outcome proportions observed in the holdout dataset. Points falling in the diagonal line represent perfect calibration of the model.

human health, could lead to significant increments in predictive performance for mortality-prognosis models, scoring amongst the highest in the available literature [5]. More analytically, the models utilizing the dispersion properties of the time series performed equally well as the ones using patterns of overall trend. While memory dynamics predicted better than sociodemographics alone, their performance was inferior to that of the other dynamics. Aggregating all the dynamical properties in one model did not seem to significantly exceed the accuracy of the models using only one of the former two. We consider that as an indication that knowledge about various aspects of medical spending captures different frailty signals but prognoses similarly.

We found that different properties, describing the underlying dynamic process of human health, could all contribute to a greater benefit in identifying high-risk individuals. The afore-stated phenomenon has not only been observed in humans but also in widely different systems. In fact, various dynamical indicators (DIORs), such as variance and lag-1 autocorrelation, have been shown to function as generic, quantitative measures of resilience status [9, 12, 34–36]. Finally, using XGBoost as our modelling algorithm, we were able to capture the same signal by using as input the raw, healthcare expenditures, without pre-specifying a number of DIORs. The latter, and based on our analysis, alludes to the fact that a flexible machine learning model that captures complex (nonlinear) interactions, can provide the same prediction benefit as one which is based on feature selection (DIORs). Nevertheless, the benefit of taking advantage of the temporality of variables, in our case healthcare expenditures, was apparent in both strategies. Thereby, we emphasize that one of the ways to improve and enrich

our prognostic toolbox is by investing in the development of datasets consisting of time series related to human health.

The decision curve analysis provided useful insights regarding the potential clinical impact of the prediction models using different threshold levels for intervention. In general, the models with dynamical indicators of healthcare expenditures were superior when treatment would be initiated at mortality estimates below 25% in the first year after spousal bereavement. Nevertheless, we do not consider that result to be prohibiting the potential use of such models in specific clinical settings given the rare occurrence of the outcome. On the contrary, we believe that our models show the greatest utility within the range of risk thresholds 0.1 through 0.2 that are plausible values for many bereaved individuals to fall within. Our analysis also showed that while our DIORs model exhibited high false negative rates, these were still greatly lower than a model prognosing mortality based on sociodemographic information, thus contributing to a more precise assessment of mortality risk after spousal loss.

We consider the sample size as well as the complete availability of all kinds of different healthcare costs in the Danish, nationwide register data, representative of the bereaved population, as major strengths of the study. We also consider the time-series based predictive framework, as novel, combining ideas from complex systems and biology, in prognostic research. To our knowledge, no other studies under the prognostic framework have explored to that extent what various dynamics of expenditures can bring to prediction of mortality. In addition, contrary to most prognostic studies which measure accuracy based on discrimination ability (AUC) [7, 37, 38], the current study assessed and internally validated those models through the lens of not only the latter, but also via calibration, overall prediction error and clinical impact, thus providing a more holistic perspective of the models' performance.

Nonetheless, the study has its limitations. The analysis has been based on spousal bereaved older adults and hence, the results might not be generalized to younger people who suffered spousal bereavement. Another potential limitation of our analysis involves the generalization to individuals outside of Denmark, since the healthcare expenditures distribution might be different in other countries of different healthcare systems which might lead to different, potentially weaker signals from the dynamical indicators. The reason for that is that Denmark has a tax-based system of healthcare, meaning that individuals who have the need to utilize healthcare services, are, under most circumstances, able to do so. That might not be the case for countries in which the healthcare system is based on out-of-pocket expenses and individuals with poor health might not be able to be captured by the healthcare expenditures distribution. In order to be able to address that argument we would need the availability of further external data to validate our models, which we did not have. In addition, our models require the availability of healthcare expenditures for individuals, so that the DIORs to be extracted. Consequently, approximately 2.5% our cohort members with no medical usage, were omitted from our analysis. While that leads to exclusion of some of the healthiest individuals, we do not consider their small fraction to be significantly changing our outcome distribution.

In conclusion, the current research provided evidence that temporal dynamics of healthcare costs have the potential to significantly improve our assessment on who is at high risk of dying after suffering spousal loss. The trajectories of medical spending (overall trend and dispersion dynamics) provided us with a higher predictive performance of mortality risk than what the baseline, sociodemographic variables could achieve. We showed evidence that a methodological approach which considers the dynamic trajectory of human health can assist in a more efficient risk profiling of bereaved individuals, which can in turn improve prognosis and support the development of advanced care planning.

## Supporting information

**S1 Fig. Calibration plot for males showing risk estimates of all-cause mortality within the year after spousal bereavement against outcome proportions observed in the holdout dataset.** Points falling in the diagonal line represent perfect calibration of the model.
(DOCX)

**S2 Fig. Calibration plot for females showing risk estimates of all-cause mortality within the year after spousal bereavement against outcome proportions observed in the holdout dataset.** Points falling in the diagonal line represent perfect calibration of the model.
(DOCX)

**S3 Fig. False negative rates of "All DIORs + Sociodemographics" model developed to estimate the risk of all-cause mortality within the year after spousal bereavement.** The x-axis for both panels A & B ('Risk Threshold') shows the range of risk threshold probabilities, that is the probabilities which when exceeded, individuals are classified as high risk of dying within 1-year. For panel A: The y-axis shows the "False Negative Rate" which is the proportion of false negatives among all positives. For panel B: The y-axis displays the "False Negative Rate Difference", that is the False Negative Rate (FNR) of "All DIORs + Sociodemographics" model minus the FNR of "Sociodemographics Only" model. The shaded purple areas around the lines of both plots represent 95% confidence intervals of the FNR and the FNR difference via bootstrapping (500 samples).
(DOCX)

**S1 Table. Performance differences of prediction models stratified on sex in the holdout set against benchmark model.**
(DOCX)

**S2 Table. TRIPOD checklist: Prediction model development and validation.**
(DOCX)

## Author Contributions

**Conceptualization:** Alexandros Katsiferis, Rudi G. J. Westendorp.

**Data curation:** Laust Hvas Mortensen.

**Formal analysis:** Alexandros Katsiferis.

**Funding acquisition:** Rudi G. J. Westendorp.

**Investigation:** Alexandros Katsiferis, Samir Bhatt, Laust Hvas Mortensen, Swapnil Mishra, Majken Karoline Jensen, Rudi G. J. Westendorp.

**Methodology:** Alexandros Katsiferis, Samir Bhatt, Laust Hvas Mortensen, Swapnil Mishra, Majken Karoline Jensen, Rudi G. J. Westendorp.

**Supervision:** Samir Bhatt, Rudi G. J. Westendorp.

**Writing – original draft:** Alexandros Katsiferis, Samir Bhatt, Laust Hvas Mortensen, Swapnil Mishra, Majken Karoline Jensen, Rudi G. J. Westendorp.

**Writing – review & editing:** Alexandros Katsiferis, Samir Bhatt, Laust Hvas Mortensen, Swapnil Mishra, Majken Karoline Jensen, Rudi G. J. Westendorp.

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
