## [Decision Letter · Decision Letter 0]

24 May 2023

PONE-D-22-31964Machine learning models of healthcare expenditures predicting mortality: A cohort study of spousal bereaved Danish individualsPLOS ONE

Dear Dr. Katsiferis,

Thank you for submitting your manuscript to PLOS ONE. After careful consideration, we feel that it has merit but does not fully meet PLOS ONE’s publication criteria as it currently stands. Therefore, we invite you to submit a revised version of the manuscript that addresses the points raised during the review process.

ACADEMIC EDITOR:This manuscript provides an in depth analysis to investigate if temporal patterns of healthcare expenditures can improve the predictive performance for mortality for individuals who loose their spouses. Overall, results indicate a pattern where healthcare expenditures improved mortality predictions the most. The following comments might help the authors with some clarity on the manuscript:

1. What method of adding predictors to the model was used?

2. How was the affluence index created?

3. Cite the Brier score and explain it as well

4. Why was it significant/important to stratify the analysis by gender

5. P-values should be included in the tables 2 and 3 or any other measure of statistical significance to observe the statistical difference between the groups. Please ensure that your decision is justified on PLOS ONE’s publication criteria and not, for example, on novelty or perceived impact.

We look forward to receiving your revised manuscript.

Kind regards,

Amna Tariq, PhD

Academic Editor

PLOS ONE

“This work is supported by The Novo Nordisk Foundation (https://novonordiskfonden.dk/) Challenge Programme for the project Harnessing the Power of Big Data to Address the Societal Challenge of Aging (NNF17OC0027812). The funders had no role in study design, data collection and analysis, decision to publish, or preparation of the manuscript.”

“This work is supported by The Novo Nordisk Foundation (https://novonordiskfonden.dk/) Challenge Programme for the project Harnessing the Power of Big Data to Address the Societal Challenge of Aging (NNF17OC0027812). The funders had no role in study design, data collection and analysis, decision to publish, or preparation of the manuscript.”

Reviewers' comments:

Reviewer's Responses to Questions

**Comments to the Author**

1. Is the manuscript technically sound, and do the data support the conclusions?

Reviewer #1: Yes

Reviewer #2: Yes

Reviewer #3: Yes

2. Has the statistical analysis been performed appropriately and rigorously? 

Reviewer #1: Yes

Reviewer #2: Yes

Reviewer #3: Yes

3. Have the authors made all data underlying the findings in their manuscript fully available?

Reviewer #1: Yes

Reviewer #2: No

Reviewer #3: Yes

4. Is the manuscript presented in an intelligible fashion and written in standard English?

Reviewer #1: Yes

Reviewer #2: Yes

Reviewer #3: Yes

5. Review Comments to the Author

Reviewer #1: Please see the comments.

• Conclusion should be clearly and concisely answered the main research question

• Briefly and precisely refer to the summary of the important results

• What was the of participation rate in the present study?

• Table 1: provide a p-value column for comparison of variables among gender subgroup

• Please explain more in detail about Affluence Index Group in methods section also insert a source as citation

• Table 1: Distribution of socio-demographic variables of the study sample should be report by Test and Trained samples

• Table 2: the column of p-value should be added.

Reviewer #2: The authors have presented a well-written and concise study of predicting mortality using machine learning models with features such as age, sex, sociodemographic factors, and healthcare expenditures. The authors conclude that adding healthcare expenditures as features to age and sex improves mortality predictability more than adding sociodemographic factors. During my review of the paper, I highlighted the following areas for improvement -

1. The authors have presented confidence intervals for AUC and Brier scores. From my understanding of the methodology, the authors have only 1 holdout set with 25% of the data. In this case, how are the confidence intervals computed since there would be only one AUC value and Brier score per model for the holdout set?

2. It would be helpful to add whether the holdout set was created using a stratified train-test split given the label imbalance (~6% mortality rate).

3. AUC is a good metric to measure model performance when the labels are fairly balanced. However, in this case, the large proportion of negative labels can result in very low false positive rates, thus affecting the AUC calculations. The author’s conclusions would be bolstered by including the comparisons of average precision (which is the area under the precision-recall curve) across the different models due to the imbalanced labels.

4. The authors have presented true positive rate in Fig 5 but did not provide the false negative rates of the models. Since the goal is to predict mortality, it would be suitable to understand the models’ performance on false negative rates. A model with high false negative rate would incorrectly predict no mortality for individuals with observed mortality and hence may be less suitable as a decision support tool.

5. Line 306 - It was not clear how raw weekly expenditures were used as features. For example, were the feature values equal to expenditure in each week starting on Jan 1, 2011? In this case, what were the feature values after an individual’s death?

6. Fig 4 - The definitions of observed mortality proportion and predicted 1-year mortality risk could not be ascertained. Please describe the axes, similar to Fig 5.

7. Line 122 - duplicated “date of”

Reviewer #3: The manuscript appears to be technically sound and the data presented support the conclusions drawn by the authors. The study is well-designed and the analysis methods used are appropriate. The authors provide a detailed description of the data and their methods, as well as the results obtained, which are presented in a clear and concise manner. The results are discussed in the context of the existing literature, and the limitations of the study are acknowledged. Overall, the manuscript appears to be a well-executed study that provides important insights into the prediction of all-cause mortality following spousal bereavement.

The research text describes a study aimed at comparing prognostic models of all-cause mortality risk based on healthcare expenditures data. The study uses Danish population registers data and analyzes healthcare expenditures of Danish citizens aged 65 years and above, who had been residents of Denmark for at least 5 years before their date of death, from January 1st, 2011, to December 31st, 2016. The study population is restricted to those who suffered spousal bereavement during the time window 2013-2016. The authors computed various Dynamic Indicator of Overall Risk (DIORs) based on different properties of the underlying dynamic process of healthcare expenditures. The DIORs were used as predictors in XGBoost tree-based models to predict all-cause mortality risk.

The research provides sufficient information on the study population, data sources, and analytical methods. The use of national population registers to obtain a large study population with a unique personal identifier for each citizen is a strength of the study.

The section describing healthcare expenditures and knowledge discovery provides a clear explanation of the computation of the DIORs.

The section describing the statistical methods is well-written and provides clear information on the XGBoost tree-based algorithm, the selection of hyperparameters, and the assessment of the models. The use of a 5-fold cross-validated dataset of the initial training set and a holdout set to validate the models' generalization ability is a strength of the study.

Overall, the study provides a clear description of the methods used to analyze the healthcare expenditures data and predict all-cause mortality risk. However, the lack of presentation of mathematical formulas for DIORs is a potential gap.

6. PLOS authors have the option to publish the peer review history of their article (what does this mean?). If published, this will include your full peer review and any attached files.

Reviewer #1: No

Reviewer #2: No

Reviewer #3: **Yes: **Vineet Gupta

---

## [Author Response · Author response to Decision Letter 0]

15 Jun 2023

See attached .docx file for response to reviewers.

---

## [Editor Report · Decision Letter 1]

24 Jul 2023

Machine learning models of healthcare expenditures predicting mortality: A cohort study of spousal bereaved Danish individuals

PONE-D-22-31964R1

Dear Dr. Katsiferis,

We’re pleased to inform you that your manuscript has been judged scientifically suitable for publication and will be formally accepted for publication once it meets all outstanding technical requirements.

Kind regards,

Amna Tariq, PhD

Academic Editor

PLOS ONE
---

## [Editor Report · Acceptance letter]

26 Jul 2023

PONE-D-22-31964R1 

Machine learning models of healthcare expenditures predicting mortality: A cohort study of spousal bereaved Danish individuals 

Dear Dr. Katsiferis:

I'm pleased to inform you that your manuscript has been deemed suitable for publication in PLOS ONE. Congratulations! Your manuscript is now with our production department. 

Kind regards, 

on behalf of

Dr. Amna Tariq 

Academic Editor

PLOS ONE